# Health literacy, cognitive ability and smoking: a cross-sectional analysis of the English Longitudinal Study of Ageing

Chloe Fawns-Ritchie,[1,2] John M Starr,[1,3] Ian J Deary[1,2]

[1]Centre for Cognitive Ageing and Cognitive Epidemiology, University of Edinburgh, Edinburgh, UK
[2]Department of Psychology, University of Edinburgh, Edinburgh, UK
[3]Alzheimer Scotland Dementia Research Centre, University of Edinburgh, Edinburgh, UK

**Correspondence to**
Professor Ian J Deary;
ian.deary@ed.ac.uk

## ABSTRACT

**Objectives** We used logistic regression to investigate whether health literacy and cognitive ability independently predicted whether participants have ever smoked and, in ever smokers, whether participants still smoked nowadays.

**Design** Cross-sectional study.

**Setting** This study used data from Wave 2 (2004–05) of the English Longitudinal Study of Ageing, which is a cohort study of adults who live in England and who, at baseline, were aged 50 years and older.

**Participants** 8734 (mean age=65.31 years, SD=10.18) English Longitudinal Study of Ageing participants who answered questions about their current and past smoking status, and completed cognitive ability and health literacy tests at Wave 2.

**Primary and secondary outcome measures** The primary outcome measures were whether participants reported ever smoking at Wave 2 and whether ever smokers reported still smoking at Wave 2.

**Results** In models adjusting for age, sex, age left full-time education and occupational social class, limited health literacy (OR=1.096, 95% CI 0.988 to 1.216) and higher general cognitive ability (OR=1.000, 95% CI 0.945 to 1.057) were not associated with reporting ever smoking. In ever smokers, limited compared with adequate health literacy was associated with greater odds of being a current smoker (OR=1.194, 95% CI 1.034 to 1.378) and a 1 SD higher general cognitive ability score was associated with reduced odds of being a current smoker (OR=0.878, 95% CI 0.810 to 0.951), when adjusting for age, sex, age left full-time education and occupational social class.

**Conclusions** When adjusting for education and occupation variables, this study found that health literacy and cognitive ability were independently associated with whether ever smokers continued to smoke nowadays, but not with whether participants had ever smoked.

## INTRODUCTION

The effects of smoking on ill health have been known for decades. The prevalence of smoking in the UK is falling and the number of smokers who are quitting is increasing.[1] Despite this, nearly 16% of the UK population were current smokers in 2016[1] and smoking

### Strengths and limitations of this study

► This study used a large sample (n=8734) from the English Longitudinal Study of Ageing, a study designed to be representative of the English population aged over 50 years.

► This analysis was cross-sectional and therefore cannot determine the direction of the association between smoking, health literacy and cognitive ability.

► This study included measures of both health literacy and cognitive ability which allowed us to investigate whether health literacy was associated with smoking status when controlling for cognitive ability.

► Smoking status was self-reported.

remains one the largest causes of preventable morbidity and mortality in the UK.[1 2] Understanding the characteristics of individuals who take up smoking and who quit smoking is important to be able to design and target smoking education and interventions.

Cognitive ability is associated with smoking. Individuals who smoke have lower scores on cognitive tests than those who have never smoked.[3–5] Smokers show steeper ageing-related cognitive decline[4–7] and have increased risk of Alzheimer's disease,[6 8] compared with non-smokers. One possible pathway between smoking and cognitive ability is that smoking has harmful consequences for the vascular system, which could in turn affect cognitive functioning.[6 9]

A perhaps complementary explanation is that individuals who have lower cognitive ability in youth are more likely to take up smoking and less likely to quit.[9–11] Corley *et al*[9] found that, when controlling for childhood cognitive ability, the association between smoking and cognitive function in old age was attenuated and, in some cases, became non-significant. Two studies[10 12] found different patterns when investigating

the relationship between childhood cognitive ability and reporting ever smoking. One study using the 1970 British Birth Cohort[10] found that individuals with higher childhood cognitive ability were less likely to have ever smoked in a sample of middle-aged participants, whereas another report, based on two of the Midspan prospective cohort studies,[12] found no association between cognitive ability in childhood and ever smoking in a sample of older adults. Both these studies, however, found that among ever smokers, individuals with higher childhood cognitive ability were more likely to quit smoking.[10 12]

A person's health literacy may also play a role in smoking status, though the evidence for an association between health literacy and smoking is mixed.[13–16] Health literacy is the capacity to acquire, process and use health information to successfully navigate all aspects of health, including the ability to use health documents, interact with healthcare professionals and undertake health-promoting behaviours to prevent future ill health.[17 18] Some studies have found that individuals with lower health literacy are more likely to smoke,[13 14] whereas others have not.[15 16] It is possible that individuals who have limited health literacy are less aware of the adverse effects of smoking on health, and may be less able to understand and use smoking cessation services.

The current study sought to determine whether health literacy and cognitive ability, when studied together, have independent associations with smoking. Drawing on the English Longitudinal Study of Ageing (ELSA), we first investigated whether health literacy and cognitive ability were independently associated with whether individuals had ever smoked. Second, we investigated whether there was a relationship between health literacy, cognitive ability and whether ever smokers continued to smoke, or quit.

## METHODS
### Participants
This study used data from ELSA, a panel study designed to be representative of individuals aged 50 years and older living in England.[19] A total of 11 391 participants took part in Wave 1 in 2002–2003. Wave 1 participants were individuals who had previously taken part in the Health Survey for England, were born before 1 March 1952 and were living in a private household in England at the first wave.[19] These participants have been followed up every 2 years, and the sample has been refreshed at subsequent waves to maintain a representative sample of participants aged over 50 years. More information on this cohort is provided elsewhere.[19] The current sample consists of participants who completed the Wave 2 (2004–2005) interview (n=8780); this is the first wave in which health literacy was assessed. Ethical approval was granted. This study conformed to the principles embodied in the Declaration of Helsinki.

### Measures
ELSA interviews were carried out using computer-assisted interviewing in the participants' own home.

### Smoking
Two aspects of smoking status (ever vs never smoker and current vs former smoker) were the outcome variables in these analyses. Participants were asked 'Have you ever smoked cigarettes?'. Participants were categorised as ever smokers if they answered 'yes' and never smokers if they answered 'no' at either Wave 1 or 2. Ever smokers were additionally asked 'Do you smoke cigarettes at all nowadays?'. Ever smokers who answered 'yes' to smoking cigarettes nowadays at Wave 2 were categorised as current smokers, whereas ever smokers who answered 'no' were categorised as former smokers.

### Health literacy
Health literacy was assessed at Wave 2 using a four-item comprehension test previously used in the International Adult Literacy Survey.[20] Participants were presented with a piece of paper containing instructions similar to those that would be found on a packet of over-the-counter medication. Participants were instructed to read the medicine label and were then asked four questions about the information on this label (eg, 'what is the maximum number of days you may take this medicine?'). The label was available to the participant to refer to at any time. This task was designed to measure the skills thought to be required to understand and use health materials correctly, such as the ability to read and use numbers in a health context.[21] One point was awarded for each correctly answered question (range 0–4). As has been done in previous ELSA studies,[22 23] health literacy scores were categorised as 'adequate' (4/4 correct) or 'limited' (≤3 correct).

### Cognitive function
Four tests of cognitive function that were administered at Wave 2 of the ELSA study were used here. These tests are thought to assess episodic memory, executive function and processing speed; these are cognitive domains which tend to decline on average with increasing age.[24 25] In the word list recall test, participants heard a list of 10 words which they had to recall immediately (immediate recall test) and again after a short delay (delayed recall test). The score on each occasion was the number of words remembered (range 0–10). Executive function was assessed using categorical verbal fluency (number of animals named in 60 s). The letter cancellation test, in which participants were to scan rows of letters and score out all Ps and Ws, was used to measure processing speed. The score is the number of Ps and Ws scored out in 60 s. Exploratory factor analysis (EFA) using principal axis factoring was used to derive a composite measure of general cognitive ability. Scores on the four cognitive tests were entered into the EFA. Prior to this, individuals who scored 0 or greater than 4 SD above the mean on the animal fluency test and the letter cancellation test

were removed. Scores of 0 indicate that the participant did not understand the task, and scores 4 SD above the mean were seen as dubiously high given the 1 min time limit for these tests. One unrotated factor was extracted which accounted for 44% of the total variance in the four cognitive tests. The loadings of the tests were: immediate word recall=0.78; delayed word recall=0.83; animal naming=0.53; letter cancellation=0.42. This factor score was converted to a z-score (mean=0.00, SD=1.00) and was used as a measure of general cognitive ability.

## Covariates

Age in years, sex, age of leaving full-time education and occupational social class were used as covariates. For confidential reasons, owing to there being few of them, participants aged over 90 years have had their age set to 90. Participants were asked at what age they left continuous full-time education (recorded as not yet finished, never went to school, 14 or under, at 15, at 16, at 17, at 18, and 19 or over). For the purpose of this study, age of leaving full-time education was categorised as 14 years or under, 15–16 years, 17–18 years and 19 years or over. Occupational social class was categorised using the National Statistics Socio-economic Classification 3 categories: managerial and professional, intermediate and routine and manual.[26]

## Patient and public involvement

Participants were not involved in the development of any part of this study.

## Statistical analysis

Two sets of analyses were carried out. First, ever smokers were compared with never smokers; second, current smokers were compared with former smokers. To determine whether ever versus never smokers and current versus former smokers differ on health literacy, general cognitive ability and sociodemographic variables, t-tests were used for normally distributed continuous variables, Mann-Whitney U tests were used for non-normal continuous variables and $\chi^2$ tests were used for categorical variables. Rank-order correlations were calculated between the predictor variables to examine any bivariate associations between these variables. Binary logistic regression was used to examine the independent associations of health literacy and general cognitive ability on smoking status. Age and sex were entered in all models. Health literacy and general cognitive ability were entered individually in models 1 and 2, respectively. To determine whether both health literacy and general cognitive ability are independently associated with smoking, both predictors were included in model 3. Model 4 additionally adjusted for age of finishing full-time education and occupational social class to determine whether any associations between health literacy, cognitive function and smoking remained after controlling for these sociodemographic variables.

## RESULTS

Of the 8780 participants who completed the Wave 2 interview, 8734 participants had complete data on smoking, cognitive ability and health literacy, and they make up the analytic sample. Participant characteristics are reported in table 1. A total of 5525 (63.3%) participants reported ever smoking, whereas 3209 (36.7%) participants reported having never smoked. Ever smokers were more likely to have limited health literacy and have a lower general cognitive ability than never smokers. Ever smokers were older, were more likely to be male, have left full-time education at a younger age and have a lower occupational social class than never smokers. A total of 1356 (15.5%) participants reported that they still smoked cigarettes at Wave 2, whereas 4169 (47.7%) participants reported that they had stopped. Current smokers were more likely to have limited health literacy than former smokers; however, the two groups did not differ on general cognitive ability. Current smokers were younger, more likely to be female, have left full-time education at a younger age and to have a lower occupational social class than former smokers. Given that current smokers were, on average, 4.5 years younger than former smokers, we tested the point-biseral correlation between smoking status and general cognitive ability, with and without controlling for age. When not controlling for age, the correlation between smoking and general cognitive ability was 0.01 (p=0.389). Adjusting for age, the correlation was −0.09 and this was significant (p<0.001).

Rank-order correlations between the predictor variables are shown in table 2. All predictor variables were significantly correlated with each other, with the exception of sex with health literacy and education. Having adequate health literacy was moderately associated with having higher general cognitive ability (r=0.31, p<0.001). Adequate health literacy was associated with having higher qualifications (r=0.23, p<0.001) and a higher occupational class (r=−0.18, p<0.001). Older adults were less likely to have adequate health literacy (r=−0.16, p<0.001). General cognitive ability was strongly correlated with age. Older individuals tended to have lower general cognitive ability (r=−0.46, p<0.001). Female participants (r=−0.10, p<0.001), individuals with higher qualifications (r=0.38, p<0.001) and higher occupational class (r=−0.25, P<0.001) tended to have higher general cognitive ability.

Table 3 shows the ORs and 95% CIs for reporting ever smoking. Adjusting for age and sex only, limited health literacy was associated with greater odds of ever smoking (model 1 OR=1.174, 95% CI 1.067 to 1.293). A 1 SD higher score in general cognitive ability was associated with an 8.1% reduction in reporting ever smoking (model 2 OR=0.919, 95% CI 0.874 to 0.967). The associations between health literacy and general cognitive ability with ever smoking remained significant, though slightly reduced in size, in the model including both health literacy and cognitive ability (model 3). In model 4, which additionally adjusted for sociodemographic variables, the associations between health literacy (OR=1.096, 95% CI

**Table 1** Participant characteristics according to smoking status (n=8734)*

| | Smoking history | | | Smoking cessation† | | |
|---|---|---|---|---|---|---|
| | Ever smoker (n=5525) | Never smoker (n=3209) | P values for difference | Current smoker (n=1356) | Former smoker (n=4169) | P values for difference |
| Health literacy, n (%) | | | 0.001 | | | <0.001 |
| Adequate | 3647 (66.0) | 2233 (69.6) | | 840 (61.9) | 2807 (67.3) | |
| Limited | 1878 (34.0) | 976 (30.4) | | 516 (38.1) | 1362 (32.7) | |
| General cognitive ability, mean (SD) | −0.04 (1.00) | 0.08 (0.99) | <0.001 | −0.02 (0.99) | −0.05 (1.01) | 0.385 |
| Age (years), mean (SD) | 65.53 (10.13) | 64.93 (10.24) | 0.005 | 62.12 (9.12) | 66.64 (10.20) | <0.001 |
| Sex, n (%) | | | 0.001 | | | <0.001 |
| Female | 2752 (49.8) | 2172 (67.7) | | 761 (56.1) | 1991 (47.8) | |
| Male | 2773 (50.2) | 1037 (32.3) | | 595 (43.9) | 2178 (52.2) | |
| Age left full-time education, n (%) | | | <0.001 | | | <0.001 |
| 14 years or under | 1104 (20.6) | 553 (17.6) | | 233 (17.7) | 871 (21.5) | |
| 15–16 years | 2936 (54.8) | 1578 (50.2) | | 856 (65.0) | 2080 (51.4) | |
| 17–18 years | 665 (12.4) | 488 (15.5) | | 128 (9.7) | 537 (13.3) | |
| 19 years or over | 657 (12.3) | 526 (16.7) | | 99 (7.5) | 558 (13.8) | |
| Occupational social class, n (%) | | | <0.001 | | | <0.001 |
| Managerial and professional | 1677 (30.8) | 1047 (33.2) | | 274 (20.6) | 1403 (34.2) | |
| Intermediate | 1263 (23.2) | 884 (28.1) | | 312 (23.4) | 951 (23.2) | |
| Routine and manual | 2499 (45.9) | 1218 (38.7) | | 747 (56.0) | 1752 (42.7) | |

*Characteristics for age left full-time education are based on a subset of 8507 participants with this data and characteristics for occupational social class are based on a subset of 8588 participants with this data.
†For smoking cessation comparisons, the ever smoker category is divided into whether ever smokers are current or former smokers.

0.988 to 1.216) and general cognitive ability (OR=1.000, 95% CI 0.945 to 1.057) with ever smoking were partly and fully attenuated, respectively, and no longer significant.

The ORs (95% CIs) for whether ever smokers reported being a current smoker at Wave 2 are shown in table 4. For this analysis, a Box-Tidwell test revealed that models violated the assumption of linearity of the logit; therefore, an age-squared term was included in these models. To overcome multicollinearity, the ORs and CIs are based on models using centred continuous variables. Controlling for age and sex only, having limited health literacy compared with adequate health literacy was associated with 49.3% greater odds of being a current smoker (model 1 OR=1.493, 95% CI 1.307 to 1.704). A 1 SD higher score in general cognitive ability was associated with 22.8% lower odds of reporting being a current smoker (model 2

**Table 2** Rank-order correlations between predictor variables (pairwise n=8367–8734)

| | Health literacy | General cognitive ability | Age (years) | Sex | Education | Occupational class |
|---|---|---|---|---|---|---|
| Health literacy | – | | | | | |
| General cognitive ability | 0.31*** | – | | | | |
| Age (years) | −0.16*** | −0.46*** | – | | | |
| Sex | 0.01 | −0.10*** | 0.02* | – | | |
| Education | 0.23*** | 0.38*** | −0.40*** | 0.00 | – | |
| Occupational class | −0.18*** | −0.25*** | 0.07*** | −0.09*** | −0.41*** | – |

*p<0.05, **p<0.01, ***p<0.001.
Health literacy was coded 0 for inadequate health literacy, 1 for adequate health literacy; sex was coded 0 for women, 1 for men; education is age left full-time education and was coded 1 for 14 years or under, 2 for 15–16 years, 3 for 17–18 years, 4 for 19 years or older; occupational social class was coded 1 for managerial and professional, 2 for intermediate, 3 for routine and manual.

**Table 3** ORs and 95% CIs from logistic regression models of whether participants have ever smoked

| | Model 1 (n=8734) | Model 2 (n=8734) | Model 3 (n=8734) | Model 4 (n=8367) |
|---|---|---|---|---|
| **Health literacy** | | | | |
| Adequate | Reference | Reference | Reference | Reference |
| Limited | 1.174 (1.067 to 1.293)** | – | 1.134 (1.026 to 1.254)* | 1.096 (0.988 to 1.216) |
| General cognitive ability† | – | 0.919 (0.874 to 0.967)** | 0.936 (0.888 to 0.987)* | 1.000 (0.945 to 1.057) |
| Age (years) | 1.004 (1.000 to 1.008) | 1.001 (0.996 to 1.006) | 1.001 (0.996 to 1.006) | 1.002 (0.996 to 1.007) |
| **Sex** | | | | |
| Female | Reference | Reference | Reference | Reference |
| Male | 2.112 (1.929 to 2.313)*** | 2.077 (1.896 to 2.276)*** | 2.087 (1.905 to 2.288)*** | 2.150 (1.955 to 2.366)*** |
| **Age left full-time education** | | | | |
| 14 years or under | – | – | – | Reference |
| 15–16 years | – | – | – | 1.016 (0.880 to 1.172) |
| 17–18 years | – | – | – | 0.828 (0.690 to 0.994)* |
| 19 years or older | – | – | – | 0.693 (0.572 to 0.839)*** |
| **Occupational class** | | | | |
| Managerial and professional | – | – | – | Reference |
| Intermediate | – | – | – | 0.919 (0.810 to 1.041) |
| Routine and manual | – | – | – | 1.204 (1.066 to 1.360)** |

*p<0.05, **p<0.01, ***p<0.001.
†ORs (95% CIs) for general cognitive ability are the odds of reporting ever smoking for a 1 SD increase in general cognitive ability.

OR=0.772, 95% CI 0.718 to 0.829). Including both health literacy and general cognitive ability in model 3 reduced the size of the associations, but they remained significant. These associations continued to remain significant, though further attenuated, in the fully adjusted model, which additionally adjusted for age completed full-time education and occupational social class (model 4 OR for limited compared with adequate health literacy=1.194, 95% CI 1.034 to 1.378; OR for a 1 SD higher score in general cognitive ability=0.878, 95% CI 0.810 to 0.951). In this final model, age left full-time education and occupational social class were also significantly associated with reporting being a current smoker. Compared with individuals who left full-time education at 14 years or under, those who left at age 17–18 or over 19 years had reduced odds of being a current smoker. Compared with those with a managerial or professional occupational class, those with a routine or manual occupational class had increased odds of being a current smoker.

## DISCUSSION

This study found that in a sample of middle-aged and older adults residing in England, health literacy and cognitive ability were independently related with whether ever smokers continue to smoke nowadays, but not with whether individuals have ever smoked. Adjusting for age and sex only, participants with limited health literacy and lower cognitive ability were more likely to report having ever smoked. However, when additionally adjusting for age left full-time education and occupational class, these associations were attenuated and became non-significant. This suggests that health literacy and cognitive function do not have associations with ever smoking that are independent of education and occupational class. In ever smokers, those with limited health literacy and poorer cognitive ability were more likely to report that they continued to smoke. These associations remained, though slightly attenuated, even after adjusting for measures of socioeconomic status.

Whereas previous studies have found associations between health literacy and smoking,[13 14] and cognitive ability and smoking,[3–10 12] to the best of our knowledge, this is the first study to find that both health literacy and cognitive function are independently associated with smoking cessation. Health literacy and cognitive function are strongly related[27–30] and some have proposed that health literacy is not a unique construct and is, rather, a subcomponent of general cognitive ability.[30] The current study, however, found that both health literacy and cognitive ability each play independent roles in predicting smoking cessation which is in support of health literacy and cognitive ability being separate, although related, constructs.

A particularly important finding from the current study was that health literacy, independent of cognitive ability, education and occupational social class, was associated with whether ever smokers continued to smoke.

**Table 4** ORs and 95% CIs from logistic regression models of whether ever smokers still smoke nowadays

|  | Model 1 (n=5525) | Model 2 (n=5525) | Model 3 (n=5525) | Model 4 (n=5280) |
|---|---|---|---|---|
| **Health literacy** |  |  |  |  |
| Adequate | Reference | Reference | Reference | Reference |
| Limited | 1.493 (1.307 to 1.704)*** | – | 1.338 (1.165 to 1.536)*** | 1.194 (1.034 to 1.378)* |
| General cognitive ability† | – | 0.772 (0.718 to 0.829)*** | 0.805 (0.747 to 0.868)*** | 0.878 (0.810 to 0.951)** |
| Age | 0.952 (0.945 to 0.958)*** | 0.943 (0.936 to 0.951)*** | 0.943 (0.936 to 0.950)*** | 0.938 (0.929 to 0.947)*** |
| Age$^2$ | 0.999 (0.999 to 1.000)** | 0.999 (0.999 to 1.000)** | 0.999 (0.999 to 1.000)** | 0.999 (0.998 to 1.000)** |
| **Sex** |  |  |  |  |
| Female | Reference | Reference | Reference | Reference |
| Male | 0.744 (0.655 to 0.845)*** | 0.707 (0.622 to 0.803)*** | 0.714 (0.628 to 0.811)*** | 0.755 (0.661 to 0.863)*** |
| **Age left full-time education** |  |  |  |  |
| 14 years or under | – | – | – | Reference |
| 15–16 years | – | – | – | 0.734 (0.593 to 0.908)** |
| 17–18 years | – | – | – | 0.515 (0.384 to 0.687)*** |
| 19 years or older | – | – | – | 0.432 (0.308 to 0.578)*** |
| **Occupational class** |  |  |  |  |
| Managerial and professional | – | – | – | Reference |
| Intermediate | – | – | – | 1.390 (1.144 to 1.689)*** |
| Routine and manual | – | – | – | 1.614 (1.375 to 1.961)*** |

*p<0.05, **p<0.01, ***p<0.001.
†ORs (95% CI) for general cognitive ability are the odds of reporting being a current smoker for a 1 SD increase in general cognitive ability.
Age $^2$, Age squared.

Health literacy, unlike already-established measures of socioeconomic status and perhaps more so than cognitive ability, is potentially modifiable.[31] Health literacy is thought to be the complex set of skills that are required to navigate all aspects of healthcare,[17 18] and include reading and numeracy skills, as well as health-related knowledge.[32] At least one component of health literacy—health knowledge—may be increased through educational programmes and interventions[32] and this in turn could lead to improved health outcomes.[33] Future research should examine whether cognitive ability and health literacy play a role in the success of smoking interventions, and should investigate whether interventions designed to increase smoking-specific health knowledge increase smoking cessation in individuals with limited health literacy.

This cross-sectional study was interested in examining the characteristics of smokers and, although a relationship between cognitive ability, health literacy and smoking cessation was identified, this study cannot determine the directionality of this association. It is possible that individuals who continue to smoke have lower cognitive ability and health literacy because smoking has damaging effects on both health literacy and cognitive ability. It is also possible that individuals who have lower cognitive ability and are less health literate are more likely to continue smoking because

they do not have the cognitive capacity or the health-related knowledge and skills required to fully comprehend the adverse effects of continuing to smoke on health, or the knowledge and skills required to access and use smoking cessation services. For cognitive ability, evidence suggests that both of these pathways may be at least partially correct. Individuals with higher cognitive ability early in life are less likely to start smoking and more likely to quit,[10 12] and smoking may cause steeper cognitive change throughout life.[4–7] A similar relationship may exist between health literacy and smoking. Further longitudinal studies which include measures of cognitive ability and health literacy in early life are needed to understand the pathways between health literacy, cognitive ability and smoking.

The key strengths of this study include the large sample size and the fact that ELSA was designed to be representative of individuals aged over 50 residing in England.[19] One limitation is that smoking status was self-reported; however, self-reported smoking has been found to be in agreement with serum cotinine measurements.[34] The smoking measures used here do not take into account the amount smoked throughout life. These results reported here may underestimate the true effect sizes because lifetime smoking was not considered.

Another limitation of this study is that the cognitive ability and health literacy tests used here were brief.

Health literacy was assessed using a four-item test that was relatively insensitive to individual differences. That is, most individuals (67.3%) answered all questions correctly. Many, more detailed, health literacy assessments are available, such as the Test of Functional Health Literacy in Adults[35 36] which is thought to be the gold-standard measure of health literacy.[37] More detailed health literacy tests may be more sensitive to detecting associations between health literacy and health. However, the brief four-item measure of health literacy used in ELSA has been found to be associated with mortality[21] and participation in cancer screening,[22] suggesting it is sensitive enough to detect associations with health.

The measure of general cognitive ability created here was constructed using a small number of brief cognitive tests that did not include, for example, reasoning that is highly loaded on general cognitive ability.[38] A better general cognitive ability measure would have been possible had more domains of cognitive function been assessed, with more detailed tests. Given other studies which have suggested that some health literacy measures are essentially aspects of cognitive function,[30 39] and given also the limited cognitive test battery used in ELSA, it is possible that some of the independent contribution of the health literacy measure here is residual cognitive capability not picked up by the limited general cognitive ability component.

In this study of middle-aged and older adults, lower cognitive ability and poorer health literacy were associated with whether ever smokers continued to smoke, even after adjusting for education and occupational class. Further research is needed to identify possible pathways between health literacy, cognitive function and starting and quitting smoking.

**Acknowledgements** We are grateful to the EnglishLongitudinal Study of Ageing participants, and to the UK Data Archive for supplying the data.

**Contributors** CF-R discussed and planned the study and analyses, analysed the data, interpreted the data and drafted the initial manuscript. JMS discussed and planned the study and analysis, interpreted the data and contributed to the manuscript. IJD discussed and planned the study and analysis, interpreted the data and contributed to the manuscript.

**Funding** This work was supported by the University of Edinburgh Centre for Cognitive Ageing and Cognitive Epidemiology, part of the cross council Lifelong Health and Wellbeing Initiative, funded by the Biotechnology and Biological Sciences Research Council (BBSRC), and Medical Research Council (MRC) (grant number MR/K026992/1).

**Competing interests** None declared.

**Patient consent** Not required.

**Ethics approval** Ethical approval for the English Longitudinal Study of Ageing was obtained from the National Research and Ethics Committee.

**Provenance and peer review** Not commissioned; externally peer reviewed.

**Data sharing statement** No additional data available. Anonymised data from the English Longitudinal Study of Ageing is available from the UK Data Service (https://www.ukdataservice.ac.uk/).

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
