## [Reviewer comments · BMJ Open]

ARTICLE DETAILS

TITLE (PROVISIONAL)	The role of health literacy and cognitive ability in smoking: a cross-sectional analysis of the English Longitudinal Study of Ageing
AUTHORS	Fawns-Ritchie, Chloe; Starr, John; Deary, Ian

VERSION 1 – REVIEW

REVIEWER	Giuseppe Gorini, Oncologic network, prevention and research Institute (ISPRO), Florence, Italy
REVIEW RETURNED	22-May-2018

GENERAL COMMENTS	This article reported in a cohort of adults aged 50 years or more that limited health literacy & lower general cognitive ability are independently associated with reporting current smoking, even after adjusting for age, education level (e.g., age left full-time education), occupational social class, and gender. Instead, reporting ever smoking is NOT associated with limited health literacy & lower cognitive ability, after adjusting for the same covariates (education level, occupational class, age, gender). Accordingly, all the manuscript must be changed. The interpretation is that smoking cessation is independently associated with adequate health literacy & higher cognitive ability, even after adjusting for age, occupational class, gender, and education level. Thus, improving health literacy through increasing health knowledge, could be important in order to promote smoking cessation within smokers with limited health literacy. 1. statistical analyses & tables: the problem is that limited health literacy & lower general cognitive ability are not associated with reporting EVER smoking, because in the model 4 of table 2 both ORs for limited health literacy & general cognitive ability were non-significant. Reporting ever smoking is associated instead only to male gender, occupational class, & education level. So, I strongly suggest to: A) please, unite tables 2 & 3 in one table reporting ONLY model 4 results for Ever and Current smokers.. It's not useful to report models without adjustments for education level & occupational class (model 3) or models adjusted only for gender, considering separately the two variables under study (models 1 & 2). B) Change abstract, main results section in the main text, and discussion in the main text, highlighting the most important results on current smoking. You can only cite in the results section of the main text & in the abstract that ever smoking ORs for limited health literacy and lower cognitive ability were significant only with no adjustments, but the main results for ever smokers is that these two variables (literacy & cognitive ability) are NOT associated to reporting ever smoking in the model 4.
--

	2. Discussion section: Directionality of association among health literacy/cognitive ability & smoking. Please, explore data (not necessarily report these analyses in the main text, but show in the letter of response to reviewers' suggestions): check the correlation i) between health literacy & "age left full-time education"; ii) cognitive ability & "age left full-time education"; iii) health literacy & occupational class; iv) cognitive ability & occupational class. I suppose that at least for health literacy, lower levels are more likely in smokers with lower education level & lower occupational class. Moreover, in order to deepen these data, you could conduct two logistic regression models in current smokers only: one with cognitive ability as dependent variable adjusting for education level, health literacy, occupational class, age, and gender; the second with health literacy as dependent variable adjusting for the other 5 variables, in order to study predictive variables of health literacy and cognitive ability . I suppose that at least for health literacy it's likely that limited health literacy may be linked to lower education level, and it reduces the self-efficacy of smoker to quit, as it happens with people with lower education level. So, for health literacy I suppose that directionality in both senses is limited. Instead, for cognitive ability, I suppose that directionality in both senses could be more probable.
--	---

REVIEWER	Raymond L Ownby, MD, PhD, MBA Nova Southeastern University, USA
REVIEW RETURNED	09-Jun-2018

GENERAL COMMENTS	BMJ Open 2018 – 023939 This is the report of an interesting and useful study that examined the extent to which health literacy and cognitive functioning are associated with smoking in a large dataset. The introduction concisely summarizes existing research, examines alternate hypotheses about a potential relation, and states a clear objective. I believe that the methods section might be improved by at least a brief description of the ELSA study and the circumstances under which it was begun and continued. Similarly, it would be helpful to the reader to know how participants in the study responded to questions, for example, on computer, on paper, or via interview. The use of principal components analysis (PCA) to create a composite cognitive ability measure simplifies the interpretation of this aspect of participants' functioning but is not really state of the art for this sort of analysis. In addition, while PCA may be appropriate for creating an index based on variables that can be assumed to be measures with little or no error (e.g., age, education, income) it is not appropriate for cognitive abilities tests that have substantial error. The resulting component loadings include elements of both true variance (that associated with the cognitive ability composite) and error variance. At a minimum, I would suggest the authors use principal axis factoring to develop their composite. This will reduce the loadings but be more precise and may even increase the power of their analysis. A more state-of-the-art analysis would be to do the full
---

	analysis in a structural equation model which could simultaneously estimate all the relevant relations among the variables. The discussion section provides a good review of the study's results and implications, especially their finding that both health literacy and cognitive ability independently are related to health behaviors. This is especially important given other authors who have suggested on the one hand that health literacy is simply literacy while on the other that it simply reflects general cognitive ability. The discussion of the study's limitations could be strengthened with additional consideration of measurement issues in both the assessments of cognition and health literacy. The composite of cognition is heavily weighted to verbal learning and memory, thus not including many components of general cognitive ability. Similarly, the measure of health literacy is not only brief but how it might be expected to be a link to smoking behavior is not clear. In the interest of full disclosure, I should point out that our group has also addressed health literacy as a combination of abilities, academic skills, and health knowledge as the authors discuss in this MS: (Ownby et al. [2014] Abilities, skills and knowledge in measures of health literacy. Patient Educ Couns. 2014 May;95(2):211-7. doi: 10.1016/j.pec.2014.02.002. Epub 2014 Feb 16).
--	--

VERSION 1 – AUTHOR RESPONSE

Reviewer 1

1. [R]eporting ever smoking is NOT associated with limited health literacy & lower cognitive ability, after adjusting for the same covariates (education level, occupational class, age, gender).

Accordingly, all the manuscript must be changed.

Response: We thank reviewer 1 for their comments. The statement in the first sentence above is correct. We now explain why we think it is important to work toward presenting that. We think it is important to highlight to the reader that health literacy and cognitive function were associated with ever smoking when only adjusting for age and sex and that these become non-significant when additionally adjusting for education and occupational class. The change in the size of the association and the fact that health literacy and cognitive function become non-significant when additionally adjusting for age left full-time education and occupational class tells us that education and occupational class are partly driving the association between health literacy and ever smoking and cognitive function and ever smoking. This is correct and very common procedure in epidemiological studies. That is, one starts with the association between two variables of interest (with basic covariates, usually age and sex)—here, that's health literacy and smoking—and then, in a planned way, ask if there are other validated variables that account for their association. To report only the fully-adjusted model would hide the initial association and obscure an important part of the study.

We agree, however, that we need to make it clear to the reader that health literacy and cognitive function do not predict ever smoking when additionally adjusting for education and occupational social class, and therefore our overall conclusion should be that cognitive function and health literacy do not predict ever smoking in the fully-adjusted model. We have therefore edited the manuscript to highlight to the reader that health literacy and cognitive ability only predict ever

smoking in models adjusting for age and sex, and not when additionally adjusting for education and occupational social class.

We now only report the fully adjusted results in the abstract.

Page 2 (Abstract): *“In models adjusting for age, sex, age left full-time education, and occupational social class, limited health literacy (OR = 1.096, 95% CI 0.988 to 1.216) and higher general cognitive ability (OR = 1.000, 95% CI 0.945 to 1.057) were not associated with reporting ever smoking. In ever smokers, limited compared to adequate health literacy was associated with greater odds of being a current smoker (OR = 1.194, 95% CI 1.034 to 1.378) and a 1SD higher general cognitive ability score was associated with reduced odds of being a current smoker (OR = 0.878, 95% CI 0.810 to 0.951), when adjusting for age, sex, age left full-time education, and occupational social class.”*

Page 3 (Abstract): *“When adjusting for education and occupation variables, this study found that health literacy and cognitive ability were independently associated with whether ever smokers continued to smoke nowadays, but not with whether participants had ever smoked.”*

In the results section, we now report the ORs and 95% CIs in the text for ever smoking model 4 (fully adjusted), to further highlight to the reader that the associations between health literacy and cognitive function become non-significant when additionally adjusting for education and occupational social class.

Page 19: *“In model 4, which additionally adjusted for sociodemographic variables, the associations between health literacy (OR = 1.096, 95% CI 0.988 to 1.216) and general cognitive ability (OR = 1.000, 95% CI 0.945 to 1.057) with ever smoking were partly and fully attenuated, respectively, and no longer significant.”*

We now make it explicitly clear in the discussion that health literacy and cognitive function only predicted ever smoking when only adjusting age and sex, and that this became non-significant in the fully adjusted model.

Page 20: *“This study found that in a sample of middle-aged and older adults residing in England, health literacy and cognitive ability were independently related with whether ever smokers continue to smoke nowadays, but not with whether individuals have ever smoked. Adjusting for age and sex only, participants with limited health literacy and lower cognitive ability were more likely to report having ever smoked. However, when additionally adjusting for age left full-time education and occupational class, these associations were attenuated and became non-significant. This suggests that health literacy and cognitive function do not have associations with ever smoking that are independent of education and occupational class.”*

Our conclusion now only details that cognitive function and health literacy were associated with smoking cessation, and not ever smoking.

Page 23: *“In this study of middle-aged and older adults, lower cognitive ability and poorer health literacy were associated with whether ever smokers continued to smoke, even after adjusting for education and occupational class.”*

We have also made some small changes to the discussion to make it clear that cognitive function and health literacy only predicted whether ever smokers continue to smoke and not ever smoking.

Page 20: *“...this is the first study to find that both health literacy and cognitive function are independently associated with smoking cessation.”*

Page 21: *“The current study, however, found that both health literacy and cognitive ability each play independent roles in predicting smoking cessation which is in support of health literacy and cognitive ability being separate, albeit related, constructs.”*

Page 21: *“This cross-sectional study was interested in examining the characteristics of smokers and, although a relationship between cognitive ability, health literacy and smoking cessation was identified, this study cannot determine the directionality of this association. It is possible that individuals who continue to smoke have lower cognitive ability and health literacy because smoking has damaging effects on both health literacy and cognitive ability. It is also possible that individuals who have lower cognitive ability and are less health literate are more likely to continue smoking because they do not have the cognitive capacity or health-related knowledge and skills*

required to fully comprehend the adverse effects of continuing to smoke on health, or the knowledge and skills required to access and use smoking cessation services.”

2. [P]lease, unite tables 2 & 3 in one table reporting ONLY model 4 results for Ever and Current smokers.. It's not useful to report models without adjustments for education level & occupational class (model 3) or models adjusted only for gender, considering separately the two variables under study (models 1 & 2).

Response: We judge that each model tells the reader important information about the association between health literacy and smoking, and between cognitive function and smoking. Adding these predictors incrementally as we have done enables us to pull apart the association between health literacy and smoking, and cognitive function and smoking. This is information the readers would not be able to see if we only reported model 4.

First, by entering only health literacy (and age and sex) in model 1 and only cognitive function (and age and sex) in model 2, we are able to compare the ORs from models 1 (health literacy only) and 2 (cognitive function only) to model 3, which includes both health literacy and cognitive function. For ever smoking, we found that the OR for health literacy reduced in size from 1.174 (95% CI 1.067 to 1.293) in model 1 to 1.134 (95% CI 1.026 to 1.254) in model 3 but remained significant. Likewise, the OR for cognitive function reduced in size from 0.919 (95% CI 0.874 to 0.967) in model 2 to 0.936 (95% CI 0.888 to 0.987) in model 3. This attenuation tells us, and readers, that the content of the health literacy and cognitive function tests overlap slightly, however, both still uniquely contribute to the prediction of whether participants have ever smoked, when not accounting for sociodemographic variables.

By introducing age left full-time education and occupational class in model 4, we can further separate out what might have been accounting for the association between health literacy and smoking, and cognitive function and smoking in model 3. For ever smoking, we find that, when additionally adjusting for education and occupational class, health literacy and cognitive function become non-significant. This tells us that education and occupational class might account for the association between health literacy and cognitive function with ever smoking. We would not know that education and occupational class contribute to the association between ever smoking and health literacy and cognitive function if we only reported model 4.

Accordingly, we have not united tables 2 and 3 into one table reporting only model 4, as suggested by the reviewer. We do however agree with the reviewer that the overall conclusion of the paper should be that health literacy and cognitive function do not predict ever smoking in models adjusting for education and occupational class, and therefore we have made changes to the manuscript to reflect and emphasise this. These changes are detailed above (see responses to reviewer 1 comment 1).

In summary, it would be incorrect to hide from the reader that there are zero-order associations between smoking variables and health literacy/cognitive ability. However, we also agree with the referee that it is important to be appropriately clear about where associations do not survive adjustment for other variables. We think the revised paper has got the balance correct.

3. Change abstract, main results section in the main text, and discussion in the main text, highlighting the most important results on current smoking. You can only cite in the results section of the main text & in the abstract that ever smoking ORs for limited health literacy and lower cognitive ability were significant only with no adjustments, but the main results for ever smokers is that these two variables (literacy & cognitive ability) are NOT associated to reporting evenr smoking in the model 4.

Response: We agree that the overall conclusion of the paper should be that health literacy and cognitive ability are not associated with reporting ever smoking when adjusting for education and occupational social class. We have therefore made changes to the manuscript to reflect this. These changes are detailed above (see responses to reviewer 1 comment 1).

4. Discussion section: Directionality of association among health literacy/cognitive ability & smoking. Please, explore data (not necessarily report these analyses in the main text, but show in the letter of response to reviewers' suggestions): check the correlation i) between health literacy & "age left full-time education"; ii) cognitive ability & "age left full-time education"; iii) health literacy & occupational class; iii) cognitive ability & occupational class. I suppose that at least for health literacy, lower levels are more likely in smokers with lower education level & lower occupational class.

Response: We have run the rank-order correlations between i) health literacy and age left full-time education ($r = 0.23, p < .001$); ii) cognitive ability and age left full-time education ($r = 0.38, p < .001$); iii) health literacy and occupational class ($r = -0.18, p < .001$); and iii) cognitive ability and occupational class ($r = -0.25, p < .001$). Thus, health literacy and cognitive function correlate with education and occupational social class.

Having calculated the correlations, as suggested by the reviewer, we think this is important that the reader is aware that many of the predictor variables correlate moderately with each other. We have therefore updated the manuscript to include a table (Table 2 of manuscript, page 14) with the rank order correlations between the predictor variables.

5. Moreover, in order to deepen these data, you could conduct two logistic regression models in current smokers only: one with cognitive ability as dependent variable adjusting for education level, health literacy, occupational class, age, and gender; the second with health literacy as dependent variable adjusting for the other 5 variables, in order to study predictive variables of health literacy and cognitive ability .

I suppose that at least for health literacy it's likely that limited health literacy may be linked to lower education level, and it reduces the self-efficacy of smoker to quit, as it happens with people with lower education level. So, for health literacy I suppose that directionality in both senses is limited. Instead, for cognitive ability, I suppose that directionality in both senses could be more probable,

Response: We have carried out two models, as requested by the reviewer. First, in current smokers only ($n = 1,294$) we ran a logistic regression with health literacy as the outcome variable and age, sex, cognitive ability, occupational class, and age left full-time education as predictors (Table 1, below). Being older, having higher general cognitive ability, and more years of education was associated with reduced odds of having inadequate health literacy (Table 1, below). Having a routine or manual occupational class, compared to managerial and professional occupational class, was associated with increased odds of having inadequate health literacy (Table 1, below).

Second, in current smokers only ($n = 1,294$) we ran a linear regression with cognitive ability as the outcome and age, sex, health literacy, occupational class, and education as the predictors (Table 2, below). Older age, being male, having inadequate compared to adequate health literacy, and having a routine or manual compared to managerial or professional occupational class was associated with having lower general cognitive ability. Having more years of education was associated with higher general cognitive ability.

Table 1: Predictors of having inadequate health literacy in current smokers only ($n = 1,294$)

	OR	95% CI		p
		Lower	Upper	
Age (years)	0.982	0.966	0.998	0.033
Sex				

Female	Reference				
Male	0.926	0.723	1.183	0.539	
General cognitive ability	0.454	0.389	0.527	< 0.001	
Occupational class					
Managerial and professional	Reference				
Intermediate	1.216	0.823	1.802	0.327	
Routine and manual	1.550	1.098	2.202	0.013	
Age left full-time education					
14 years or under	Reference				
15-16 years	0.703	0.487	1.015	0.060	
17-18 years	0.559	0.321	0.962	0.038	
19 years or over	0.645	0.342	1.192	0.168	

Table 2: Predictors of general cognitive ability in current smokers only (n = 1,294)

	beta	SE	p
Age (years)	-0.037	0.003	< 0.001
Sex			
Female	Reference		
Male	-0.212	0.046	< 0.001
Health literacy			
Adequate	Reference		
Inadequate	-0.542	0.048	< 0.001
Occupational class			
Managerial and professional	Reference		
Intermediate	-0.101	0.070	0.151
Routine and manual	-0.379	0.063	< 0.001
Age left full-time education			
14 years or under	Reference		
15-16 years	0.090	0.072	0.207

17-18 years			
	0.220	0.101	0.030
19 years or over			< 0.001
	0.429	0.112	

The models above tell us that health literacy and cognitive ability are associated with these sociodemographic variables, and provide further understanding of the attenuation seen for health literacy and cognitive function between model 3 (without education and occupational class) and model 4 (with education and occupational class), i.e., that is it probably due to the associations between health literacy and cognitive ability with education and occupational class.

As the reviewer suggests, limited health literacy is related to having had fewer years of education. However, we control for education (and occupational class) in the fully-adjusted model and health literacy (and cognitive function) still predicts whether ever smokers continue to smoke nowadays. What is especially interesting about model 4 (fully-adjusted model) is that health literacy and cognitive ability remain significant predictors, though slightly attenuated, when additionally controlling for education and occupational class. That is, health literacy and cognitive function are partly independent of each other, and uniquely contribute to the prediction of whether ever smokers continue to smoke nowadays, even when controlling for education and occupational class.

We stress that this is a cross-sectional study and therefore we cannot disentangle the directionality of the association between health literacy, cognitive ability and current smoking. We think it is important to make it clear to the reader that these results only tell us that there is an association between health literacy and cognitive function with whether ever smokers continue to smoke nowadays. We cannot determine whether smoking causes individuals to have poorer health literacy and cognitive function or whether individuals who had poorer cognitive function and health literacy to begin with are more likely to continue to smoke. The revised paper has the following text:

Page 21: *“This cross-sectional study was interested in examining the characteristics of smokers and, although a relationship between cognitive ability, health literacy and smoking cessation was identified, this study cannot determine the directionality of this association. It is possible that individuals who continue to smoke have lower cognitive ability and health literacy because smoking has damaging effects on both health literacy and cognitive ability. It is also possible that individuals who have lower cognitive ability and are less health literate are more likely to continue smoking because they do not have the cognitive capacity or health-related knowledge and skills required to fully comprehend the adverse effects of continuing to smoke on health, or the knowledge and skills required to access and use smoking cessation services.”*

Reviewer 2

1. I believe that the methods section might be improved by at least a brief description of the ELSA study and the circumstances under which it was begun and continued.

Response: We thank reviewer 2 for their comments. We have now updated the methods section of the manuscript to provide more detail about the ELSA study including how it began and subsequent follow-up.

Pages 6-7: *“This study used data from ELSA, a panel study designed to be representative of individuals aged 50 years and older living in England.[19] A total of 11,391 participants took part in Wave 1 in 2002-03. Wave 1 participants were individuals who had previously taken part in the Health Survey for England, were born before 1 March 1952, and were living in a private household in England at the first wave.[19] These participants have been followed up every two years, and the sample has been refreshed at subsequent waves to maintain a representative sample of participants aged over 50 years. More information on this cohort is provided elsewhere.[19]”*

2. [I]t would be helpful to the reader to know how participants in the study responded to questions, for example, on computer, on paper, or via interview.

Response: We have now added a sentence to the methods to detail that interviews took place in the participants' own home using computer assisted interviewing.

Page 7: *“ELSA interviews were carried out using computer-assisted interviewing in the participants own home.”*

3. The use of principal components analysis (PCA) to create a composite cognitive ability measure simplifies the interpretation of this aspect of participants' functioning but is not really state of the art for this sort of analysis. In addition, while PCA may be appropriate for creating an index based on variables that can be assumed to be measures with little or no error (e.g., age, education, income) it is not appropriate for cognitive abilities tests that have substantial error. The resulting component loadings include elements of both true variance (that associated with the cognitive ability composite) and error variance.

At a minimum, I would suggest the authors use principal axis factoring to develop their composite. This will reduce the loadings but be more precise and may even increase the power of their analysis. A more state-of-the-art analysis would be to do the full analysis in a structural equation model which could simultaneously estimate all the relevant relations among the variables.

Response: We used PCA originally because it is a useful and widely-used method for generating a composite score of general cognitive ability from a number of correlated tests. For example, PCA was used by us and our collaborators from over 50 additional cohorts to create the composite measure of general cognitive function used in the CHARGE consortium to investigate the genetics of cognitive function (Davies et al. 2015. *Mol Psychiatry*;20:183-192. doi:10.1038/mp.2014.188).

As requested by the reviewer, we created a composite of general cognitive ability using principal axis factoring (PFA) and we have re-run the analyses using this new composite. The methods and results (see changes to Tables 1, 3 and 4) have been changed to reflect the use of this new composite measure.

Page 8: *"Exploratory factor analysis (EFA) using principal axis factoring was used to derive a composite measure of general cognitive ability. Scores on the four cognitive tests were entered into the EFA."*

Pages 8-9: *"One unrotated factor was extracted which accounted for 44% of the total variance in the four cognitive tests. The loadings of the tests were: immediate word recall = 0.78; delayed word recall = 0.83; animal naming = 0.53; letter cancellation = 0.42. This factor score was converted to a z-score (mean = 0.00, SD = 1.00) and was used as a measure of general cognitive ability."*

As stated by the reviewer, the loadings for the four cognitive tests are slightly lower using the PFA-derived composite than when we used the PCA-derived composite. This is predictable, of course, because PCA uses all the variance in the tests (with 1s in the leading diagonal), whereas PFA uses only the shared variance (with a measure of communality) in the leading diagonal. However, it is universally found with cognitive tests that a regression-based score from PCA correlates almost perfectly with a regression-based score from principal axis factoring. This is exactly what we found. The correlation between the PAF-derived general cognitive ability measure made for this revision and the PCA-derived general cognitive ability measure was 0.98. Thus, the two measures are almost identical. The results reported using the PAF-derived general cognitive ability score in this revision are almost identical to the results reported when we used PCA to create this composite in the original version of this manuscript.

4. The discussion of the study's limitations could be strengthened with additional consideration of measurement issues in both the assessments of cognition and health literacy. The composite of cognition is heavily weighted to verbal learning and memory, thus not including many components of general cognitive ability. Similarly, the measure of health literacy is not only brief but how it might be expected to be a link to smoking behavior is not clear.

Response: We agree that we should make readers aware of the limitations of the general cognitive ability and health literacy measures. We have therefore edited the discussion to detail that the measure of general cognitive ability created here was created using a limited number of brief tests.

Page 23: *“The measure of general cognitive ability created here was constructed using a small number of brief cognitive tests that did not include, for example, reasoning that is highly loaded on general cognitive ability.[38] A better general cognitive ability measure would have been possible had more domains of cognitive function been assessed, with more detailed tests. Given other studies which have suggested that some health literacy measures are essentially aspects of cognitive function,[30, 39] and given also the limited cognitive test battery used in ELSA, it is possible that some of the independent contribution of the health literacy measure here is residual cognitive capability not picked up by the limited general cognitive ability component.”*

While the health literacy measure in ELSA is a brief four-item test that suffers from a ceiling effect (67.3% of participants score the highest score), this measure has been found to be associated with health outcomes (mortality; Bostock & Steptoe. *BMJ* 2012;344:e1602. doi:10.1136/bmj.e1602.) and health behaviours (colorectal cancer screening; Kobayashi, Wardle & Wagner. *Prev Med* 2014;61:100-5. doi:10.1016/j.ypmed.2013.11.012). We have edited the discussion to highlight that this test has been found to be sensitive enough to detect associations with health.

Page 22: *“Health literacy was assessed using a four-item test that was relatively insensitive to individual differences. That is, most individuals (67.3%) answered all questions correctly. Many, more detailed, health literacy assessments are available, such as the Test of Functional Health Literacy in Adults[35, 36] which is thought to be the gold-standard measure of health literacy.[37] More detailed health literacy tests may be more sensitive to detecting associations between health literacy and health. However, the brief four-item measure of health literacy used in ELSA has been found to be associated with mortality[21] and participation in cancer screening,[22] suggesting it is sensitive enough to detect associations with health.”*

We have also edited the discussion to detail how having inadequate health literacy (and poorer cognitive ability) may be linked to smoking behaviour.

Page 21 *“It is possible that individuals who continue to smoke have lower cognitive ability and health literacy because smoking has damaging effects on both health literacy and cognitive ability. It is also possible that individuals who have lower cognitive ability and are less health literate are more likely to continue smoking because they do not have the cognitive capacity or health-related knowledge and skills required to fully comprehend the adverse effects of continuing to smoke on health, or the knowledge and skills required to access and use smoking cessation services.”*

We, however, also highlight that our study is cross-sectional and therefore we cannot determine the directionality of the associations:

Page 21: *“This cross-sectional study was interested in examining the characteristics of smokers and, although a relationship between cognitive ability, health literacy and smoking cessation was identified, this study cannot determine the directionality of this association.”*

VERSION 2 – REVIEW

REVIEWER	Giuseppe Gorini ISPRO, Florence, Italy
REVIEW RETURNED	31-Jul-2018
GENERAL COMMENTS	Authors fully answered to all the questions raised by the two revieweres and the Editor. The article improved a lot after this revision. I think that it is ready to be published.